# Maternal hypertensive disorder of pregnancy and offspring early-onset cardiovascular disease in childhood, adolescence, and young adulthood: A national population-based cohort study

Chen Huang[1‡], Jiong Li[2,3‡], Guoyou Qin[1‡], Zeyan Liew[4,5], Jing Hu[1], Krisztina D. László[6], Fangbiao Tao[7,8], Carsten Obel[9], Jørn Olsen[3], Yongfu Yu[1,3]*

1 Department of Biostatistics, School of Public Health, and The Key Laboratory of Public Health Safety of Ministry of Education, Fudan University, Shanghai, China, 2 Ministry of Education-Shanghai Key Laboratory of Children's Environmental Health, Xinhua Hospital, Shanghai Jiao Tong University School of Medicine, Shanghai, China, 3 Department of Clinical Medicine—Department of Clinical Epidemiology, Aarhus University, Aarhus, Denmark, 4 Department of Environmental Health Sciences, Yale School of Public Health, New Haven, Connecticut, United States of America, 5 Yale Center for Perinatal, Pediatric, and Environmental Epidemiology, Yale School of Public, New Haven, Connecticut, United States of America, 6 Department of Global Public Health, Karolinska Institutet, Stockholm, Sweden, 7 Department of Maternal, Child and Adolescent Health, School of Public Health, Anhui Medical University, Hefei, China, 8 Anhui Provincial Key Laboratory of Population Health & Aristogenics, Hefei, China, 9 Section for General Medical Practice, Department of Public Health, Aarhus University, Aarhus, Denmark

‡ These authors share first authorship on this work.
* yu@fudan.edu.cn, yoyu@clin.au.dk

**Data Availability Statement:** All data is stored at the secure platform of Denmark Statistics, which is the central authority on Danish statistics with the

## Abstract

### Background

The prevalence of cardiovascular disease (CVD) has been increasing in children, adolescents, and young adults in recent decades. Exposure to adverse intrauterine environment in fetal life may contribute to the elevated risk of early-onset CVD. Many studies have shown that maternal hypertensive disorders of pregnancy (HDP) are associated with increased risks of congenital heart disease, high blood pressure, increased BMI, and systemic vascular dysfunction in offspring. However, empirical evidence on the association between prenatal exposure to maternal HDP and early-onset CVD in childhood and adolescence remains limited.

### Methods and findings

We conducted a population-based cohort study using Danish national health registers, including 2,491,340 individuals born in Denmark from 1977 to 2018. Follow-up started at birth and ended at the first diagnosis of CVD, emigration, death, or 31 December 2018, whichever came first. Exposure of maternal HDP was categorized as preeclampsia or eclampsia ($n = 68,387$), gestational hypertension ($n = 18,603$), and pregestational hypertension ($n = 15,062$). Outcome was the diagnosis of early-onset CVD from birth to young

mission to collect, compile and publish statistics on the Danish society. Due to restrictions related to Danish law and protecting patient privacy, the combined set of data as used in this study can only be made available through a trusted third party, Statistics Denmark (https://www.dst.dk/en/kontakt). This state organisation holds the data used for this study. University-based Danish scientific organisations can be authorized to work with data within Statistics Denmark and such organisation can provide access to individual scientists inside and outside of Denmark. Researchers can apply for access to these data when the request is approved by the Danish Data Protection Agency: https://www.datatilsynet.dk, the email address for the Danish Data Protection Agency is: dt@datatilsynet.dk. Requests for data may be sent to Statistics Denmark: http://www.dst.dk/en/OmDS/organisation/TelefonbogOrg.aspx?kontor=13&tlfbogsort=sektion or the Danish Data Protection Agency: https://www.datatilsynet.dk.

**Funding:** This study was supported by a grant from Shanghai Rising-Star Program (21QA1401300) to YY; grants from the Independent Research Fund Denmark (DFF-6110-00019B, DFF-9039-00010B, and DFF-1030-00012B) to JL; a grant from the Nordic Cancer Union (R275-A15770) to JL; a grant from the Karen Elise Jensens Fond (2016) to JL; a grant from Novo Nordisk Foundation (NNF18OC0052029) to JL; grants from the National Natural Science Foundation of China (82073570 and 11871164 to GQ); and a grant from Swedish Heart and Lung Foundation (20180306) to KL. The funders had no role in study design, data collection and analysis, decision to publish, or preparation of the manuscript.

**Competing interests:** I have read the journal's policy and the authors of this manuscript have the following competing interests: during the past five years KL received research grants from the Swedish Council of Working Life and Social Research, Heart and Lung Foundation, Karolinska Institutet Research Foundation, Clas Groschinsky Memorial Foundation and the Swedish Society of Medicine.

**Abbreviations:** CVD, cardiovascular disease; DNPR, Danish National Patient Register; HDP, hypertensive disorders of pregnancy; HELLP, hemolysis, elevated liver enzymes, and low platelet; HR, hazard ratio; ICD, International Classification of Diseases.

adulthood (up to 40 years old). We performed Cox proportional hazards regression to evaluate the associations and whether the association differed by maternal history of CVD or diabetes before childbirth. We further assessed the association by timing of onset and severity of preeclampsia. The median follow-up time was 18.37 years, and 51.3% of the participants were males. A total of 4,532 offspring in the exposed group (2.47 per 1,000 person-years) and 94,457 in the unexposed group (2.03 per 1,000 person-years) were diagnosed with CVD. We found that exposure to maternal HDP was associated with an increased risk of early-onset CVD (hazard ratio [HR]: 1.23; 95% CI = 1.19 to 1.26; $P < 0.001$). The HRs for preeclampsia or eclampsia, gestational hypertension, and pregestational hypertension were 1.22 (95% CI, 1.18 to 1.26; $P < 0.001$), 1.25 (95% CI, 1.17 to 1.34; $P < 0.001$), and 1.28 (95% CI, 1.15 to 1.42; $P < 0.001$), respectively. We also observed increased risks for type-specific CVDs, in particular for hypertensive disease (HR, 2.11; 95% CI, 1.96 to 2.27; $P < 0.001$) and myocardial infarction (HR, 1.49; 95% CI, 1.12 to 1.98; $P = 0.007$). Strong associations were found among offspring of mothers with CVD history (HR, 1.67; 95% CI, 1.41 to 1.98; $P < 0.001$) or comorbid diabetes (HR, 1.56; 95% CI, 1.34 to 1.83; $P < 0.001$). When considering timing of onset and severity of preeclampsia on offspring CVD, the strongest association was observed for early-onset and severe preeclampsia (HR, 1.48, 95% CI, 1.30 to 1.67; $P < 0.001$). Study limitations include the lack of information on certain potential confounders (including smoking, physical activity, and alcohol consumption) and limited generalizability in other countries with varying disparities in healthcare.

## Conclusions

Offspring born to mothers with HDP, especially mothers with CVD or diabetes history, were at increased risks of overall and certain type-specific early-onset CVDs in their first decades of life. Further research is warranted to better understand the mechanisms underlying the relationship between maternal HDP and early-onset CVD in offspring.

## Author summary

### Why was this study done?

- The prevalence of cardiovascular disease (CVD) has been increasing in children, adolescents, and young adults in recent decades in developed countries.

- Maternal hypertensive disorders of pregnancy (HDP) is associated with an increased risk of congenital heart disease and a number of risk factors of CVD in offspring.

- Little is known about whether and to what extent prenatal exposure to HDP affects the development of early-onset CVD in offspring from birth to adolescence and beyond.

## What did the researchers do and find?

- We conducted a population-based cohort study that included all 2,491,340 live births in Denmark from 1977 to 2018 and followed them from birth to early adulthood (up to 40 years).

- We found that individuals born to mothers with HDP had a 23% increased risk of early-onset CVD in offspring, especially of those mothers with a history of CVD (67% increased risk) or diabetes (56% increased risk).

## What do these findings mean?

- Offspring born to mothers with HDP, especially mothers with CVD or diabetes, are at an increased risk of early-onset CVD from birth to early adulthood.

- These findings suggest that better management of maternal HDP, particularly in early phase of pregnancy, may improve cardiovascular health of children and adolescents and beyond, in terms of reducing the risk of early-onset CVD.

- Further research is warranted to better understand the mechanisms underlying the relationship between maternal HDP and early-onset CVD in offspring in early decades of life.

## Introduction

Cardiovascular disease (CVD) remains one of the leading causes of death worldwide [1,2], with a rising prevalence of CVD in children, adolescents, and young adults over the past few decades in developed countries and many undeveloped countries [3,4]. In addition to conventional risk factors of CVD, such as obesity, physical inactivity, dyslipidemia [3,5,6], Barker's fetal origin theory proposed that CVD may have a prenatal origin [7–9]. An increasing body of evidence has suggested intergenerational associations between maternal illness during pregnancy and the risk of CVD in offspring [4,10–12].

Hypertensive disorders of pregnancy (HDP), including preeclampsia, eclampsia, gestational hypertension, and pregestational hypertension, complicates about 3% to 10% of pregnancies and has also been increasing in recent decades [13–15]. Empirical evidence has shown that children born to mothers with HDP had increased risks of congenital heart disease, high blood pressure, increased BMI, and systemic vascular dysfunction [16–20]. Previous studies have suggested that pregnancies complicated by HDP may lead to long-term changes in cardiac and vascular functions in offspring through fetal programming, which could, in turn, increase the risk of CVD in offspring later in life [7–9,17]. Although there has been some evidence suggesting a higher risk of stroke and hypertension in offspring with maternal HDP [21–25], little is known about whether or to what extent prenatal exposure to maternal HDP would increase the risk of overall and type-specific CVDs in the first decades of life.

Using the Danish national health registries, we aimed to examine the association between maternal HDP and early-onset CVD in offspring from birth to young adulthood (up to 40 years) and whether coexisting maternal history of CVD and diabetes further increased the risk

of CVD among offspring [4]. We also assessed whether those associations differed by timing of onset and severity of preeclampsia [21,26].

## Methods

### Ethics statement

The study was approved by the Data Protection Agency (Record No. 2013-41-2569). By Danish law, no informed consent is required for a register-based study based on anonymized data.

### Study design and participants

A unique civil personal identification number is assigned to all residents in Denmark, which allows individual-level linkage across various national registries (detailed descriptions of registers are provided in S2 Text) [27,28]. We conducted a population-based cohort study including all live births in Denmark between 1977 and 2018 (*N* = 2,537,421). The final cohort comprised 2,491,340 individuals after excluding 46,081 children with diagnosed congenital heart disease (Fig 1). The follow-up started at birth and ended at the first diagnosis of CVD, emigration, death, or 31 December 2018, whichever came first. The detailed prespecified study protocol is available in S1 Text. We have followed the Strengthening the Reporting of Observational Studies in Epidemiology (STROBE) guidelines (STROBE Checklist is provided in S1 Checklist).

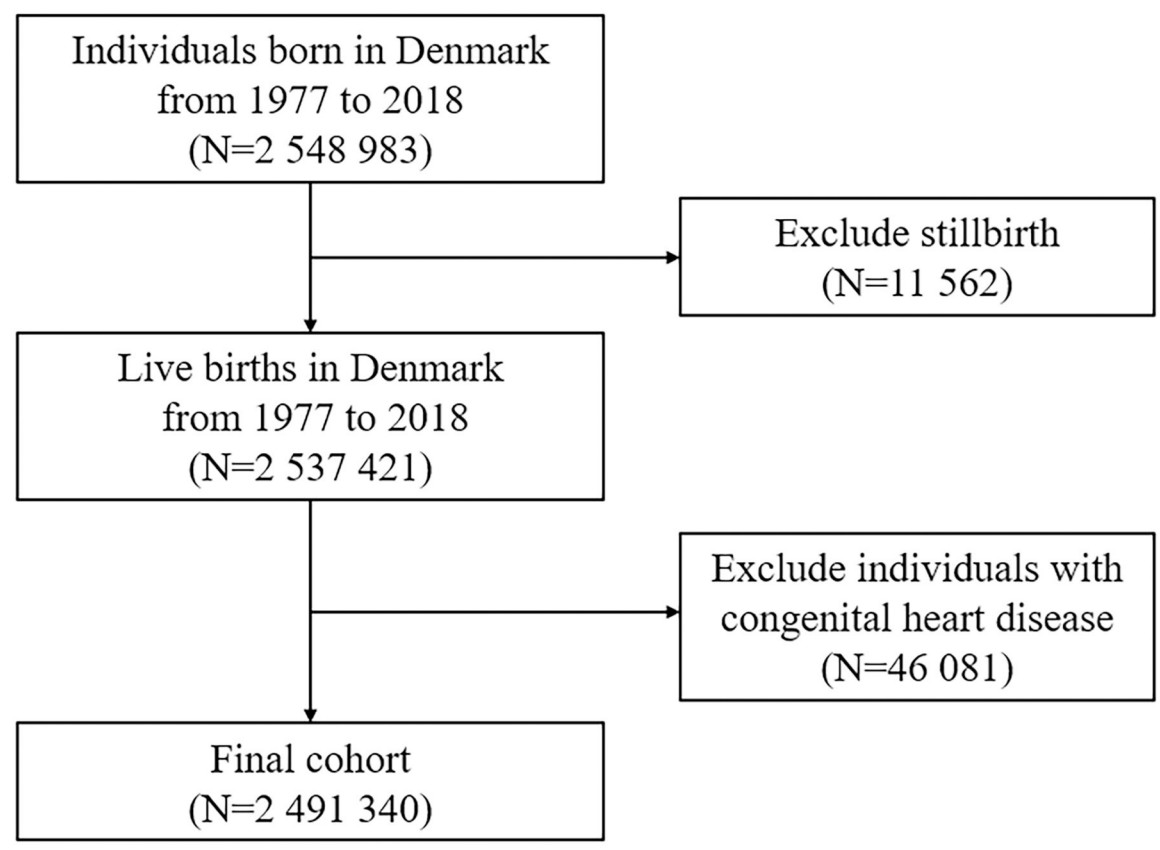

**Fig 1. Flow chart of study population.**

## Maternal hypertensive disorders of pregnancy

Information on maternal HDP was retrieved from the Danish National Patient Register (DNPR) [27,28], using the International Classification of Diseases (ICD; ICD-8, 1978 to 1993; ICD-10, 1994 and forward) (S1 Table). HDP was classified as (1) preeclampsia or eclampsia; (2) gestational hypertension; and (3) pregestational hypertension. Preeclampsia was further categorized into unspecified preeclampsia, moderate preeclampsia, severe preeclampsia, and hemolysis, elevated liver enzymes, and low platelet (HELLP) syndrome according to the severity. For women who had more than one diagnosis of HDP, we categorized them according to the hierarchy: eclampsia, preeclampsia, pregestational hypertension, and gestational hypertension.

Preeclampsia was further categorized into early-onset preeclampsia (diagnosed before 34 weeks of gestation) and late-onset preeclampsia (diagnosed at or after 34 weeks of gestation) [26]. According to the severity of preeclampsia, preeclampsia was also categorized into moderate preeclampsia and severe preeclampsia (including severe preeclampsia and the HELLP syndrome).

## Outcome of interest

The outcome of interest was early-onset CVD (excluding congenital heart disease), defined as the first occurrence of CVD in the DNPR and the Danish Cause of Death Register (Diagnostic codes and surgical codes for CVD were provided in S2 Table) [27,28]. We further investigated type-specific CVDs, such as myocardial infarction, cerebrovascular disease, stroke, heart failure, atrial fibrillation, hypertensive disease, deep vein thrombosis, pulmonary embolism, rheumatic heart disease, and peripheral arterial disease.

## Covariates

Potential confounders were selected by directed acyclic graphs (S1 Fig), including sex (male, female), singleton (yes, no), birth year of the child (1977 to 1980, 5-year intervals during 1981 to 2015, and 2016 to 2018), maternal age (<20, 20 to 24, 25 to 29, 30 to 34, or ≥35 years), maternal education (0 to 9, 10 to 14, or ≥15 years), maternal income at birth (no income, 3 tertiles), maternal prepregnancy BMI (underweight <18.5, normal 18.5 to 24.9, overweight 25.0 to 29.9, obese ≥30.0), maternal smoking during pregnancy (yes or no), parity (1, 2, or ≥3 children), maternal cohabitation (single or cohabitating), maternal residence (Copenhagen, cities with ≥100,000 inhabitants, or other), maternal history of diabetes, and maternal and parental history of CVD before childbirth (yes or no). A missing indicator method was used to deal with missing values. A detailed description of the covariates is presented in S3 Text.

## Statistical analysis

Considering non-CVD deaths as the competing events, competing risk analysis was performed to estimate cumulative incidence of CVD among offspring exposed and unexposed to maternal HDP. We used Cox regression to estimate hazard ratios (HRs) and 95% CIs to assess the association between maternal HDP and overall or type-specific CVD in offspring. The proportional hazards assumption was assessed graphically using the log-minus-log plot, suggesting that there was no obvious violation. We examined the interaction term between maternal HDP and maternal history of CVD or diabetes to assess whether the association was varied by maternal CVD or diabetes. Besides, we assessed the association by timing of onset and severity of preeclampsia (moderate, severe eclampsia, and HELLP syndrome).

We performed the following sensitivity analyses: (1) In order to assess the influence of family or genetic factors, we conducted sibship analysis by restricting offspring to sibling pairs born to same mother but different father (half-sibling) or same father and mother (full-sibling) to compare the difference in the outcomes of each sibling exposed to maternal HDP and the unexposed sibling. (2) We evaluated whether timing of delivery would affect the observed associations by dividing offspring to preterm birth and term birth. (3) We undertook stratified analysis by baseline characteristics including offspring sex, singleton, parity, maternal age, maternal education, maternal smoking during pregnancy, maternal cohabitation, maternal residence, maternal history of diabetes, and maternal and parental history of CVD before childbirth. (4) We used paternal hypertension before pregnancy as "control exposure" to examine the underlying genetic or family factors of the association. (5) We assessed the association between maternal HDP and CVD in offspring according to the timing of diagnosis of maternal HDP since childbirth (diagnosed before childbirth and diagnosed ≤3 years, 3 to 5 years, 5 to 10 years, and 10 to 15 years after childbirth). (6) We performed subanalyses: further adjusted for paternal hypertension; due to the change in ICD code and availability of data on confounders, the main analyses restricted to offspring born after 1991, 1994, and 2004; multiple imputation and complete cases analyses. All analyses were performed using SAS 9.4 (SAS Institute, Cary, North Carolina) and Stata 15.1 (StataCorp, College Station, Texas, United States).

## Results

Among 2,491,340 live-born offspring in the final cohort, 102,052 (4.10%) individuals were exposed to maternal HDP (preeclampsia or eclampsia: 2.74%; gestational hypertension: 0.75%; pregestational hypertension: 0.60%). A total of 88,275 offspring (3.55%) were censored during the follow-up, of which 68,675 (2.76%) were due to emigration and 19,600 (0.79%) were due to noncardiovascular death. Mothers with HDP were more likely to be primiparous women with lower education, live alone, and to have a history of diabetes or CVD. Offspring exposed to maternal HDP also had a higher proportion with parental history of CVD (Table 1).

During a follow-up of up to 40 years (median: 18.37 years, IQR: 9.13 to 27.28 years), 4,532 offspring (2.47 per 1,000 person-years) were diagnosed with CVD in the exposed cohort and 94,457 (2.03 per 1,000 person-years) in the unexposed cohort. Offspring exposed to maternal HDP had a higher risk of developing CVD in their first 40 years of life, compared with offspring without maternal HDP (Fig 2). Maternal HDP was associated with 23% increased risk of early-onset CVD in offspring (HR: 1.23; 95% CI: 1.19 to 1.26; $P < 0.001$) in the fully adjusted model. The risk of early-onset CVD was higher among offspring exposed to preeclampsia and eclampsia (1.22, 95% CI: 1.18 to 1.26; $P < 0.001$), gestational hypertension (HR, 1.25; 95% CI, 1.17 to 1.34; $P < 0.001$), and pregestational hypertension (HR, 1.28; 95% CI, 1.15 to 1.42; $P < 0.001$), respectively, compared to offspring of mothers without HDP. We also observed increased risks for most type-specific CVDs, in particular hypertensive disease (HR, 2.11; 95% CI, 1.96 to 2.27; $P < 0.001$), myocardial infarction (HR, 1.49; 95% CI, 1.12 to 1.98; $P = 0.007$), pulmonary embolism (HR, 1.33; 95% CI, 1.11 to 1.58; $P = 0.002$), and heart failure (HR, 1.30; 95% CI, 1.02 to 1.66; $P = 0.037$) (Table 2).

We found offspring of mothers with both HDP and history of CVD had a higher risk of early-onset CVD (HR, 1.67; 95% CI, 1.41 to 1.98; $P < 0.001$), compared to offspring born to mothers with HDP alone (HR, 1.23; 95% CI, 1.19 to 1.26; $P < 0.001$). Offspring born to mothers with HDP and history of diabetes also tended to have a higher risk of early-onset CVD (HR, 1.56; 95% CI, 1.34 to 1.83; $P < 0.001$), compared to offspring of mothers with HDP alone (HR, 1.23; 95% CI, 1.19 to 1.27; $P < 0.001$) (Table 3).

**Table 1. Baseline characteristics according to offspring's exposure to maternal HDP, Denmark, 1977–2018.**

| Characteristics[a] | No HDP (n = 2,389,288) | Preeclampsia or eclampsia[b] (n = 68,387) | Pregestational hypertension (n = 15,062) | Gestational hypertension (n = 18,603) | Total (n = 2,491,340) |
|---|---|---|---|---|---|
| **Singleton** | | | | | |
| No | 73,123 (3.1) | 6,142 (9.0) | 633 (4.2) | 791 (4.3) | 80,689 (3.2) |
| Yes | 2,316,165 (96.9) | 62,245 (91.0) | 14,429 (95.8) | 17,812 (95.7) | 2,410,651 (96.8) |
| **Sex** | | | | | |
| Boy | 1,224,718 (51.3) | 35,566 (52.0) | 7,766 (51.6) | 9,656 (51.9) | 1,277,706 (51.3) |
| Girl | 1,163,239 (48.7) | 32,782 (47.9) | 7,295 (48.4) | 8,938 (48.0) | 1,212,254 (48.7) |
| Unknown | 1,331 (0.1) | 39 (0.1) | 1 (0.0) | 9 (0.0) | 1,380 (0.1) |
| **Maternal parity** | | | | | |
| 1 | 1,061,709 (44.4) | 44,903 (65.7) | 4,980 (33.1) | 10,910 (58.6) | 1,122,502 (45.1) |
| 2 | 895,651 (37.5) | 15,903 (23.3) | 6,380 (42.4) | 5,012 (26.9) | 922,946 (37.0) |
| ≥3 | 431,928 (18.1) | 7,581 (11.1) | 3,702 (24.6) | 2,681 (14.4) | 445,892 (17.9) |
| **Maternal age at childbirth (years)** | | | | | |
| <20 | 54,569 (2.3) | 2,051 (3.0) | 46 (0.3) | 270 (1.5) | 56,936 (2.3) |
| 20–24 | 413,120 (17.3) | 13,999 (20.5) | 935 (6.2) | 2,685 (14.4) | 430,739 (17.3) |
| 25–29 | 868,600 (36.4) | 24,488 (35.8) | 3,684 (24.5) | 6,136 (33.0) | 902,908 (36.2) |
| 30–34 | 722,932 (30.3) | 17,864 (26.1) | 5,586 (37.1) | 5,690 (30.6) | 752,072 (30.2) |
| 35+ | 330,067 (13.8) | 9,985 (14.6) | 4,811 (31.9) | 3,822 (20.5) | 348,685 (14.0) |
| **Maternal smoking during pregnancy[c]** | | | | | |
| No | 1,306,384 (77.3) | 38,415 (81.0) | 11,744 (83.8) | 11,536 (84.2) | 1,368,079 (77.5) |
| Yes | 310,460 (18.4) | 6,684 (14.1) | 1,722 (12.3) | 1,709 (12.5) | 320,575 (18.2) |
| Unknown | 74,034 (4.4) | 2,304 (4.9) | 552 (3.9) | 461 (3.4) | 77,351 (4.4) |
| **Maternal education at childbirth, years** | | | | | |
| 0–9 | 620,503 (26.0) | 19,272 (28.2) | 2,627 (17.4) | 4,157 (22.3) | 646,559 (26.0) |
| 10–14 | 1,016,818 (42.6) | 30,354 (44.4) | 6,629 (44.0) | 8,315 (44.7) | 1,062,116 (42.6) |
| 15+ | 708,837 (29.7) | 17,972 (26.3) | 5,661 (37.6) | 5,890 (31.7) | 738,360 (29.6) |
| Unknown | 43,130 (1.8) | 789 (1.2) | 145 (1.0) | 241 (1.3) | 44,305 (1.8) |
| **Maternal cohabitation at childbirth** | | | | | |
| No | 1,084,136 (45.4) | 35,198 (51.5) | 6,724 (44.6) | 8,974 (48.2) | 1,135,032 (45.6) |
| Yes | 1,301,473 (54.5) | 33,167 (48.5) | 8,336 (55.3) | 9,622 (51.7) | 1,352,598 (54.3) |
| Unknown | 3,679 (0.2) | 22 (0.0) | 2 (0.0) | 7 (0.0) | 3,710 (0.1) |
| **Maternal residence at childbirth** | | | | | |
| Copenhagen | 277,438 (11.6) | 7,690 (11.2) | 1,572 (10.4) | 2,106 (11.3) | 288,806 (11.6) |
| Big cities ≥100,000 inhabitants | 306,552 (12.8) | 9,181 (13.4) | 1,959 (13) | 3,025 (16.3) | 320,717 (12.9) |
| Others | 1,805,298 (75.6) | 51,516 (75.3) | 11,531 (76.6) | 13,472 (72.4) | 1,881,817 (75.5) |
| **Maternal CVD history before childbirth** | | | | | |
| No | 2,327,447 (97.4) | 66,150 (96.7) | 13,937 (92.5) | 17,956 (96.5) | 2,425,490 (97.4) |
| Yes | 61,841 (2.6) | 2,237 (3.3) | 1,125 (7.5) | 647 (3.5) | 65,850 (2.6) |
| **Paternal CVD history before birth of the child** | | | | | |
| No | 2,286,742 (95.7) | 65,082 (95.2) | 14,136 (93.9) | 17,672 (95) | 2,383,632 (95.7) |
| Yes | 77,890 (3.3) | 2,368 (3.5) | 746 (5) | 692 (3.7) | 81,696 (3.3) |

(*Continued*)

**Table 1.** (Continued)

| Characteristics[a] | No HDP (n = 2,389,288) | Preeclampsia or eclampsia[b] (n = 68,387) | Pregestational hypertension (n = 15,062) | Gestational hypertension (n = 18,603) | Total (n = 2,491,340) |
|---|---|---|---|---|---|
| Unknown | 24,656 (1.0) | 937 (1.4) | 180 (1.2) | 239 (1.3) | 26,012 (1.0) |
| **Maternal DM history before childbirth** | | | | | |
| No | 2,350,874 (98.4) | 65,529 (95.8) | 13,719 (91.1) | 17,603 (94.6) | 2,447,725 (98.2) |
| Yes | 38,414 (1.6) | 2,858 (4.2) | 1,343 (8.9) | 1,000 (5.4) | 43,615 (1.8) |

aHR, adjusted hazard ratio; cHR, crude hazard ratio; CVD, cardiovascular disease; DM, diabetes mellitus; HDP, hypertensive disorders of pregnancy; HELLP, hemolysis, elevated liver enzymes, and low platelet.

[a]Expressed as frequency (percentage).

[b]Includes all preeclampsia or eclampsia diagnoses (moderate preeclampsia, severe preeclampsia, HELLP syndrome, unspecified preeclampsia, and eclampsia).

[c]Maternal smoking during pregnancy was available from 1991 to 2018.

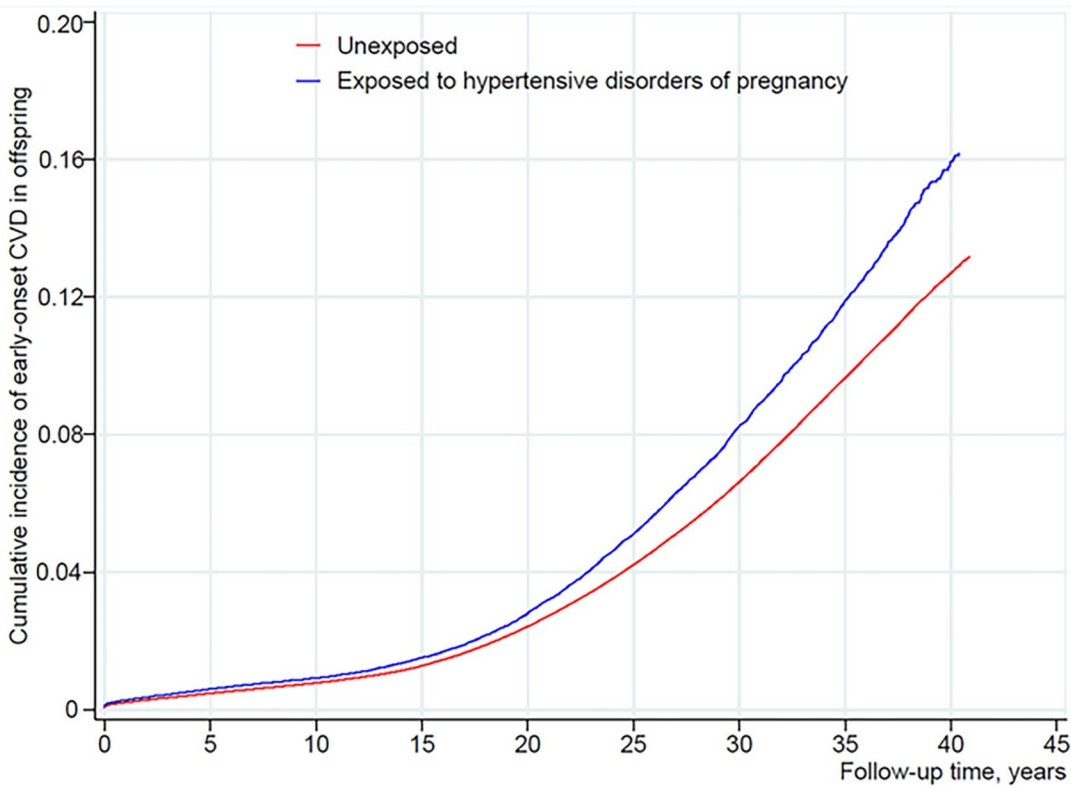

**Number at risk**

| | | | | | | | | | | |
|---|---|---|---|---|---|---|---|---|---|---|
| Unexposed | 2389288 | 2063438 | 1776881 | 1471497 | 1157628 | 827520 | 520605 | 273851 | 48487 | 0 |
| Exposed | 102052 | 85171 | 69519 | 56111 | 43726 | 31509 | 20229 | 10240 | 1639 | 0 |

**Fig 2. Cumulative incidence of early-onset CVD among offspring exposed and unexposed to HDP.** CVD, cardiovascular disease; HDP, hypertensive disorders of pregnancy.

**Table 2. HRs for associations between maternal HDP and overall early-onset CVD and type-specific CVDs in offspring.**

| Outcome[a] | Exposure | No. of CVD cases | Rate (1/10³) | cHR (95% CI) | P value | aHR[b] (95% CI) | P value |
|---|---|---|---|---|---|---|---|
| **Overall CVD** | **No maternal HDP** | 94,457 | 2.03 | 1.0(Reference) | | 1.0(Reference) | |
| | **Maternal HDP** | 4,532 | 2.47 | 1.24 (1.21–1.28) | <0.001 | 1.23 (1.19–1.26) | <0.001 |
| | **Preeclampsia or eclampsia** | 3,372 | 2.52 | 1.23 (1.19–1.27) | <0.001 | 1.22 (1.18–1.26) | <0.001 |
| | Preeclampsia | 3,345 | 2.53 | 1.23 (1.19–1.27) | <0.001 | 1.22 (1.18–1.26) | <0.001 |
| | Moderate | 2,607 | 2.55 | 1.21 (1.16–1.26) | <0.001 | 1.21 (1.16–1.25) | <0.001 |
| | Severe | 502 | 2.50 | 1.35 (1.24–1.48) | <0.001 | 1.31 (1.20–1.43) | <0.001 |
| | HELLP syndrome | 30 | 1.73 | 1.73 (1.21–2.47) | 0.003 | 1.38 (0.97–1.98) | 0.077 |
| | Unspecified | 206 | 2.44 | 1.16 (1.01–1.33) | 0.031 | 1.16 (1.01–1.32) | 0.039 |
| | Eclampsia | 27 | 2.13 | 1.09 (0.75–1.59) | 0.653 | 1.07 (0.73–1.56) | 0.728 |
| | **Hypertension** | 1,160 | 2.32 | 1.29 (1.22–1.37) | <0.001 | 1.26 (1.19–1.33) | <0.001 |
| | Pregestational | 351 | 1.97 | 1.44 (1.30–1.60) | <0.001 | 1.28 (1.15–1.42) | <0.001 |
| | Gestational | 809 | 2.51 | 1.24 (1.16–1.33) | <0.001 | 1.25 (1.17–1.34) | <0.001 |
| **Specific CVD** | | | | | | | |
| **Myocardial infarction** | No maternal HDP | 867 | 0.02 | 1.0(Reference) | | 1.0(Reference) | |
| | Maternal HDP | 50 | 0.03 | 1.50 (1.13–2.00) | 0.005 | 1.49 (1.12–1.98) | 0.007 |
| | Preeclampsia or eclampsia | 41 | 0.03 | 1.61 (1.17–2.20) | 0.003 | 1.56 (1.14–2.14) | 0.006 |
| | Hypertension | 9 | 0.02 | 1.17 (0.61–2.25) | 0.643 | 1.23 (0.64–2.37) | 0.538 |
| **Cerebrovascular disease** | No maternal HDP | 6,618 | 0.14 | 1.0(Reference) | | 1.0(Reference) | |
| | Maternal HDP | 317 | 0.17 | 1.22 (1.09–1.37) | <0.001 | 1.20 (1.07–1.35) | 0.002 |
| | Preeclampsia or eclampsia | 246 | 0.18 | 1.27 (1.12–1.44) | <0.001 | 1.24 (1.09–1.41) | <0.001 |
| | Hypertension | 71 | 0.14 | 1.09 (0.86–1.37) | 0.488 | 1.09 (0.86–1.37) | 0.491 |
| **Stroke** | No maternal HDP | 4,107 | 0.09 | 1.0(Reference) | | 1.0(Reference) | |
| | Maternal HDP | 209 | 0.11 | 1.29 (1.13–1.49) | <0.001 | 1.26 (1.10–1.45) | 0.001 |
| | Preeclampsia or eclampsia | 164 | 0.12 | 1.37 (1.17–1.60) | <0.001 | 1.32 (1.13–1.55) | <0.001 |
| | Hypertension | 45 | 0.09 | 1.08 (0.81–1.45) | 0.593 | 1.08 (0.80–1.45) | 0.623 |
| **Heart failure** | No maternal HDP | 1,321 | 0.03 | 1.0(Reference) | | 1.0(Reference) | |
| | Maternal HDP | 68 | 0.04 | 1.32 (1.03–1.68) | 0.027 | 1.30 (1.02–1.66) | 0.037 |
| | Preeclampsia or eclampsia | 53 | 0.04 | 1.37 (1.04–1.80) | 0.024 | 1.34 (1.01–1.76) | 0.040 |
| | Hypertension | 15 | 0.03 | 1.15 (0.69–1.92) | 0.582 | 1.18 (0.71–1.97) | 0.517 |
| **Atrial fibrillation** | No maternal HDP | 2,461 | 0.05 | 1.0(Reference) | | 1.0(Reference) | |
| | Maternal HDP | 110 | 0.06 | 1.16 (0.96–1.41) | 0.122 | 1.16 (0.95–1.40) | 0.140 |
| | Preeclampsia or eclampsia | 93 | 0.07 | 1.28 (1.04–1.57) | 0.020 | 1.26 (1.03–1.56) | 0.027 |
| | Hypertension | 17 | 0.03 | 0.78 (0.48–1.25) | 0.305 | 0.79 (0.49–1.27) | 0.325 |
| **Hypertensive disease** | No maternal HDP | 9,892 | 0.21 | 1.0(Reference) | | 1.0(Reference) | |
| | Maternal HDP | 822 | 0.44 | 2.17 (2.02–2.33) | <0.001 | 2.11 (1.96–2.27) | <0.001 |
| | Preeclampsia or eclampsia | 577 | 0.42 | 1.99 (1.83–2.16) | <0.001 | 1.94 (1.78–2.11) | <0.001 |
| | Hypertension | 245 | 0.48 | 2.76 (2.43–3.13) | <0.001 | 2.67 (2.35–3.03) | <0.001 |
| **Deep vein thrombosis** | No maternal HDP | 5,084 | 0.11 | 1.0(Reference) | | 1.0(Reference) | |
| | Maternal HDP | 223 | 0.12 | 1.14 (1.00–1.31) | 0.050 | 1.14 (1.00–1.30) | 0.056 |
| | Preeclampsia or eclampsia | 178 | 0.13 | 1.18 (1.02–1.37) | 0.028 | 1.16 (1.00–1.35) | 0.047 |
| | Hypertension | 45 | 0.09 | 1.01 (0.76–1.36) | 0.934 | 1.06 (0.79–1.42) | 0.720 |
| **Pulmonary embolism** | No maternal HDP | 2,577 | 0.05 | 1.0(Reference) | | 1.0(Reference) | |
| | Maternal HDP | 132 | 0.07 | 1.33 (1.12–1.59) | 0.001 | 1.33 (1.11–1.58) | 0.002 |
| | Preeclampsia or eclampsia | 99 | 0.07 | 1.30 (1.06–1.59) | 0.011 | 1.27 (1.04–1.56) | 0.019 |
| | Hypertension | 33 | 0.06 | 1.45 (1.03–2.04) | 0.035 | 1.51 (1.07–2.13) | 0.018 |
| **Rheumatic heart disease** | No maternal HDP | 302 | 0.01 | 1.0(Reference) | | 1.0(Reference) | |
| | Maternal HDP | 13 | 0.01 | 1.09 (0.63–1.90) | 0.763 | 1.13 (0.65–1.98) | 0.659 |

(*Continued*)

**Table 2.** (Continued)

| Outcome[a] | Exposure | No. of CVD cases | Rate (1/10³) | cHR (95% CI) | P value | aHR[b] (95% CI) | P value |
|---|---|---|---|---|---|---|---|
| | Preeclampsia or eclampsia[c] | - | - | - | - | - | - |
| | Hypertension[c] | - | - | - | - | - | - |
| **Peripheral arterial disease** | No maternal HDP | 511 | 0.01 | 1.0(Reference) | | 1.0(Reference) | |
| | Maternal HDP | 26 | 0.01 | 1.32 (0.89–1.96) | 0.168 | 1.31 (0.88–1.94) | 0.183 |
| | Preeclampsia or eclampsia | 19 | 0.01 | 1.27 (0.80–2.00) | 0.314 | 1.24 (0.78–1.97) | 0.355 |
| | Hypertension | 7 | 0.01 | 1.49 (0.71–3.14) | 0.295 | 1.53 (0.72–3.23) | 0.266 |

aHR, adjusted hazard ratio; cHR, crude hazard ratio; CVD, cardiovascular disease; HDP, hypertensive disorders of pregnancy; HELLP, hemolysis, elevated liver enzymes, and low platelet; HR, hazard ratio; ICD, International Classification of Diseases.

[a]Overall CVD (ICD-8: 390 to 444.1, 444.3 to 458, 782.4; ICD-10: I00 to I99). Myocardial infarction (ICD-8: 410; ICD-10: I21), cerebrovascular disease (ICD-8: 430 to 438; ICD-10: I60 to I69), stroke (ICD-8: 430 to 436; ICD-10: I61 to I64), heart failure (ICD-8: 427.0, 427.1, 782.4; ICD-10: I110, I130, I132, I50), atrial fibrillation (ICD-8: 427.93, 427.94; ICD-10: I48), hypertensive disease (ICD-8: 400 to 404; ICD-10: I10 to I15), deep vein thrombosis (ICD-8: 451.00; ICD-10: I80.1 to I80.3), pulmonary embolism (ICD-8: 450.99; ICD-10: I26), rheumatic heart disease (ICD-8: 393 to 398; ICD-10: I05 to I09), and peripheral arterial disease (ICD-8: 443.89 to 443.99; ICD-10: I73.9).

[b]Adjusted for calendar year, sex, singleton status, parity, maternal age, maternal smoking, maternal education, maternal cohabitation, maternal country of origin, maternal income at birth, maternal BMI, maternal residence at birth, maternal history of CVD and diabetes before childbirth, and paternal history of CVD before childbirth.

[c]<6 cases are not allowed to report due to data protection in Denmark.

Offspring born to mothers with early-onset preeclampsia had a higher risk of early-onset CVD (HR, 1.30; 95% CI, 1.22 to 1.39; $P < 0.001$), compared with late-onset preeclampsia (HR, 1.19; 95% CI, 1.14 to 1.24; $P < 0.001$). The risk of early-onset CVD tended to increase with the severity of preeclampsia, the estimated risk for severe preeclampsia and HELLP syndrome (HR, 1.32, 95% CI, 1.21 to 1.43; $P < 0.001$) was higher than moderate preeclampsia (HR, 1.21; 95% CI, 1.16 to 1.25; $P < 0.001$). Considering both timing of onset and severity of

**Table 3. The joint effect of maternal HDP and maternal CVD/maternal diabetes history before childbirth on early-onset CVD in offspring.**

| Attributing effects | No. of CVD cases | Rate (1/10³) | cHR (95% CI) | P value | aHR[a] (95% CI) | P value |
|---|---|---|---|---|---|---|
| **Interaction for HDP and maternal CVD history** | | | | | | |
| Main effects | | | | | | |
| Maternal HDP only | 4,397 | 2.46 | 1.24 (1.20–1.28) | <0.001 | 1.23 (1.19–1.26) | <0.001 |
| Maternal CVD only | 1,906 | 2.24 | 1.37 (1.31–1.43) | <0.001 | 1.29 (1.24–1.35) | <0.001 |
| Joint effects | | | | | | |
| Maternal HDP and CVD | 135 | 2.83 | 1.83 (1.54–2.16) | <0.001 | 1.67 (1.41–1.98) | <0.001 |
| **Interaction for HDP and maternal diabetes history** | | | | | | |
| Main effects | | | | | | |
| Maternal HDP only | 4,376 | 2.46 | 1.23 (1.20–1.27) | <0.001 | 1.23 (1.19–1.27) | <0.001 |
| Maternal diabetes only | 929 | 2.10 | 1.37 (1.28–1.46) | <0.001 | 1.26 (1.18–1.34) | <0.001 |
| Joint effects | | | | | | |
| Maternal HDP and diabetes | 156 | 2.62 | 1.70 (1.45–1.99) | <0.001 | 1.56 (1.34–1.83) | <0.001 |

aHR, adjusted hazard ratio; cHR, crude hazard ratio; CVD, cardiovascular disease; HDP, hypertensive disorders of pregnancy.

[a]Adjusted for calendar year, sex, singleton status, parity, maternal age, maternal smoking, maternal education, maternal cohabitation, maternal country of origin, maternal income at birth, maternal BMI, maternal residence at birth, maternal history of CVD and diabetes before childbirth, and paternal history of CVD before childbirth.

**Table 4. The risk of early-onset CVD in offspring according to the timing and severity of maternal preeclampsia.**

|  | No. of CVD cases | Rate (1/10³) | cHR (95% CI) | P value | aHRᵃ (95% CI) | P value |
|---|---|---|---|---|---|---|
| **By timing of preeclampsiaᵇ** |  |  |  |  |  |  |
| Early-onset | 904 | 2.52 | 1.37 (1.28–1.46) | <0.001 | 1.30 (1.22–1.39) | <0.001 |
| Late-onset | 2,235 | 2.54 | 1.19 (1.14–1.24) | <0.001 | 1.19 (1.14–1.24) | <0.001 |
| **By severity of preeclampsia** |  |  |  |  |  |  |
| Moderate | 2,607 | 2.55 | 1.21 (1.16–1.26) | <0.001 | 1.21 (1.16–1.25) | <0.001 |
| Severe and HELLP | 532 | 2.44 | 1.37 (1.26–1.49) | <0.001 | 1.32 (1.21–1.43) | <0.001 |
| **Timing and severity of preeclampsia** |  |  |  |  |  |  |
| Late-onset* Moderate | 1,953 | 2.58 | 1.18 (1.13–1.23) | <0.001 | 1.19 (1.14–1.25) | <0.001 |
| Late-onset* Severe/HELLP | 282 | 2.31 | 1.23 (1.09–1.38) | <0.001 | 1.20 (1.07–1.35) | 0.002 |
| Early-onset* Moderate | 654 | 2.49 | 1.30 (1.21–1.41) | <0.001 | 1.25 (1.16–1.35) | <0.001 |
| Early-onset* Severe/HELLP | 250 | 2.61 | 1.57 (1.39–1.78) | <0.001 | 1.48 (1.30–1.67) | <0.001 |

aHR, adjusted hazard ratio; cHR, crude hazard ratio; CVD, cardiovascular disease; HELLP, hemolysis, elevated liver enzymes, and low platelet.

ᵃAdjusted for calendar year, sex, singleton status, parity, maternal age, maternal smoking, maternal education, maternal cohabitation, maternal country of origin, maternal income at birth, maternal BMI, maternal residence at birth, maternal history of CVD and diabetes before childbirth, and paternal history of CVD before childbirth.

ᵇIncludes moderate preeclampsia, severe preeclampsia, and HELLP syndrome.

preeclampsia on offspring CVD, the strongest association was found for early-onset and severe preeclampsia (HR, 1.48; 95% CI, 1.30 to 1.67; P < 0.001) (Table 4).

Sibship analyses restricting offspring to sibling pairs with same mother but different father (half-sibling) or sibling pairs with same mother and father (full-sibling) showed the increased risks of most type-specific CVDs (S2 Fig), such as hypertensive disease (half-sibling [HR, 2.05; 95% CI, 1.88 to 2.24; P < 0.001]; full-sibling [HR, 2.08; 95% CI, 1.89 to 2.28; P < 0.001]), pulmonary embolism (half-sibling [HR, 1.47; 95% CI, 1.20 to 1.79; P < 0.001]; full-sibling [HR, 1.41; 95% CI, 1.13 to 1.75]; P = 0.002), and deep vein thrombosis (half-sibling [HR, 1.28; 95% CI, 1.10 to 1.49; P = 0.001]; full-sibling [HR, 1.31; 95% CI, 1.11 to 1.54; P = 0.001]). Analyses using paternal hypertension before pregnancy as "control exposure" indicated a weak association (HR, 1.07; 95% CI, 0.95 to 1.22; P = 0.267) between paternal hypertension and offspring CVD (S3 Table). Moreover, for the timing of the diagnosis of maternal HDP, the association was the strongest when maternal HDP was diagnosed before childbirth (HR, 1.20; 95% CI, 1.16 to 1.23; P < 0.001). The associations attenuated with elapsed time after birth when maternal HDP diagnosis was made (S3 Fig). The analyses stratified by preterm births or baseline characteristics, additionally adjusted for paternal hypertension, restricted to offspring born after 1991, 1994, 2001, used multiple imputation and complete cases analyses, yielded similar results to those obtained in the primary analyses (S4–S6 Tables).

## Discussion

In this large population-based cohort study with a follow-up of up to 40 years, we found that offspring born to mothers with preeclampsia or eclampsia, gestational hypertension, and pregestational hypertension had 22%, 25%, and 28% increased risks of early-onset CVD in offspring from birth to early adulthood, respectively, compared to offspring born to mothers without HDP. Similar associations were observed in certain specific types of CVD, for example, hypertensive disease and myocardial infarction. Stronger associations were found among offspring of mothers with a history of diabetes (56% increased risk) or CVD (67% increased

risk). Timing of onset and severity of preeclampsia would also influence the association, and the strongest association was observed for early-onset and severe preeclampsia.

Multiple case–control and cohort studies have provided evidence of the association between HDP and a range of CVD risk factors and CVD-related diseases in offspring during neonatal period, childhood, adolescence, and young adulthood, including biochemical markers of CVD in newborns (lower birth weight and smaller abdominal circumference) [29], higher systolic and diastolic blood pressure [19,23,30–33], BMI [19,31,34], and waist circumference [18], unfavorable lipid profile [18,35], and obesity [36]. There has been limited empirical evidence on the associations of HDP (mainly preeclampsia) with CVD morbidity and few subtypes of CVD, including stroke and hypertension [21–25]. A population-based study of offspring up to 18 years of age in Israel found that severe preeclampsia was associated with more than 2-fold increased risk of cardiovascular morbidity (including hypertension, arrhythmia, and heart failure) in offspring born at term, but not in offspring born preterm [21]. Studies from New England Birth Cohort and Western Australian Pregnancy Cohort found that young adults exposed to maternal HDP was at an increased risk of self-reported hypertension [22,23]. The Helsinki Birth Cohort Study demonstrated that the risks of thrombotic stroke and hypertension were higher among offspring exposed to mothers with gestational hypertension and severe preeclampsia [24]. The empirical evidence on the association remains preliminary, due to the relatively small sample size or short follow-up that did not permit detailed analyses for subtypes of exposure and outcomes. The validity of self-reported diseases might also be prone to bias [21–24]. Our large cohort study found an increased risk of overall and certain type-specific CVDs from birth to young adulthood (up to 40 years old) in individuals of mothers with preeclampsia or eclampsia, which was in line with previous studies. And we also observed similar increased risks in relation to prenatal exposure to maternal gestational hypertension and pregestational hypertension. In addition to increased risks of hypertensive disease, heart failure, and stroke that were observed in previous studies, we provided evidence on the association of maternal HDP with several other types of CVD like myocardial infarction for the first time. The differences in the association between maternal HDP and type-specific CVDs in offspring may be due to complex pathophysiology and the effects of various future risk factors for the development of type-specific CVDs [37]. Further investigation on the underlying mechanisms and to explore the effects of other different risk factors during life for specific CVDs are warranted. We further observed an increasing trend of CVD risk in offspring with increased severity of preeclampsia, consistent with the observation in a previous study [24]. Interestingly, we observed an increased risk early-onset CVD in offspring born to mothers with preeclampsia, irrespective of being preterm or not, suggesting that the association between preeclampsia and early-onset CVD in offspring may be independent of preterm birth or gestational age [21].

Several underlying mechanisms may be used to interpret our findings. It has been proposed that in utero exposure to adverse intrauterine environment was associated with a series of cardiovascular outcomes later in life [7–9]. HDP may exert an adverse effect on abnormal placental development in early pregnancy, which would lead to an ischemic and hypoxic environment for fetal development from the first trimester and activate an overexpress of anti-angiogenic factors from the second trimester, thereby inhibiting vascular endothelial and placental growth [15,17]. Placental ischemia and intrauterine hypoxia environment would result in impaired metabolism, ventricular and myocardial hypoplasia, and epicardial detachment in rat fetuses [38,39]. These abnormal intrauterine environmental factors would affect cardiac development later in life by inducing adverse structural and functional changes to the cardiovascular system both in fetal and postfetal life [7–9,17,40–42]. Several studies have found that adverse structural and functional changes in the heart and blood vessels in offspring born to mothers with preeclampsia, including systemic vascular dysfunction, decreased measures of

microvascular function, and smaller hearts from childhood [16,42,43]. In addition to the abovementioned mechanisms, damaged DNA and epigenetic changes, an overactive sympathetic nervous system, shared genetic and environmental characteristics, and lifestyle factors may contribute to the association between HDP and CVD in offspring [17].

We found higher risks of CVD in offspring born to mothers with both HDP and a history of diabetes or CVD, compared to offspring born to mothers with no HDP and no history of diabetes or CVD. Although the pathophysiology and interplay of maternal HDP and maternal history of CVD with diabetes on the development of CVD in offspring remains less understood, the added influence of maternal history of diabetes or CVD on CVD risk in offspring needs further research to evaluate the burden of multimorbidity during pregnancy.

A previous study has reported that severe preeclampsia was reported to be an independent risk factor for cardiovascular morbidity in offspring [21]. It was suggested that placental gene expression between severe early-onset and late-onset preeclampsia was different and that placentas in the early preeclampsia groups had a higher risk of infarction [26,44]. In line with these evidences, we observed that offspring born to mothers with early-onset and severe preeclampsia had a higher risk of developing CVD.

## Strengths and limitations

This study has some strengths. First, the prospectively collected register data and the inclusion of all Danish live-born children minimized the probability of recall bias and selection bias. Second, the long-term follow-up and the large sample size allowed us to investigate the association between HDP and the CVD subtypes from birth to childhood, adolescence, and beyond. Third, we were able to use sibship design to assess the influences of uncontrolled confounding due to shared inheritance or common characteristics within the family.

Some limitations are also worth mentioning. First, we could not exclude the possibility of residual confounding due to lack of information on certain important confounders, such as smoking status, physical activity, alcohol use, diet, and other lifestyle factors [2,45]. However, we have adjusted for a large number of potential confounding factors, which have been considered as the most important ones. Moreover, sibling-matched analysis yielded similar results. In addition, the considerably great impact of maternal hypertension, compared with paternal hypertension, on the CVD risk in offspring, further suggested that observed associations are unlikely to be attributable completely to uncontrolled confounding. Second, there might be misclassification in the diagnosis of HDP and CVD. However, in a validity study of preeclampsia-related diagnosis in Denmark, a moderate sensitivity of 69% and a high specificity of 99% were shown for all-type preeclampsia [15,46]. Besides, the diagnoses of the most common CVD were recorded accurately, and the positive predictive values exceeded 90% in DNPR [47]. Third, our study was conducted in Denmark where a secure social welfare system has well been established [27], thus our findings may not be generalized to other countries. Further studies are warranted to replicate our findings in developing countries in particular, where prevalence of maternal HDP and early-onset CVD might be different from the countries in the Nordic setting.

## Conclusions

Our findings suggest that offspring born to mothers with HDP, especially mothers with CVD history or diabetes history, are at increased risks of overall and certain type-specific early-onset CVDs in their first decades of life. Further research is warranted to better understand the mechanisms underlying the relationship between maternal HDP and early-onset CVD in offspring.

## Supporting information

**S1 Checklist. STROBE checklist for reporting cohort studies.**
(DOCX)

**S1 Text. Study protocol.**
(DOCX)

**S2 Text. Detailed description of registers used in this study.**
(DOCX)

**S3 Text. Detailed description of covariates.**
(DOCX)

**S1 Table. Exposure classification of hypertensive disorders from the International Classification of Diseases, the eighth and 10th version (ICD-8 and ICD-10).**
(DOCX)

**S2 Table. Outcome classification of overall CVD and specific CVD from the International Classification of Diseases, the eighth and 10th version (ICD-8 and ICD-10).**
(DOCX)

**S3 Table. Associations between paternal hypertension before pregnancy and early-onset CVD in offspring.**
(DOCX)

**S4 Table. Associations between maternal preeclampsia or eclampsia and early-onset CVD in offspring according to the timing of the delivery.**
(DOCX)

**S5 Table. Associations between maternal hypertensive disorder of pregnancy and early-onset CVD in offspring, by characteristics.**
(DOCX)

**S6 Table. Subanalyses of the association between maternal hypertensive disorder of pregnancy and early-onset CVD in offspring.**
(DOCX)

**S1 Fig. Causal diagram showing selection of covariates for confounding control.**
(DOCX)

**S2 Fig. Associations between maternal hypertensive disorders of pregnancy and early-onset CVD in offspring of sibling pairs.**
(DOCX)

**S3 Fig. Associations between maternal hypertensive disorder of pregnancy and early-onset CVD in offspring, according to the timing of the maternal HDP diagnosis.**
(DOCX)

**S4 Fig. The log-minus-log survival curve.**
(DOCX)

## Author Contributions

**Conceptualization:** Jiong Li, Guoyou Qin, Yongfu Yu.

**Data curation:** Jiong Li, Yongfu Yu.

**Formal analysis:** Chen Huang, Jiong Li, Guoyou Qin, Zeyan Liew, Jing Hu, Krisztina D. László, Fangbiao Tao, Carsten Obel, Jørn Olsen, Yongfu Yu.

**Funding acquisition:** Jiong Li, Guoyou Qin, Krisztina D. László, Yongfu Yu.

**Investigation:** Jiong Li, Guoyou Qin, Yongfu Yu.

**Methodology:** Chen Huang, Jiong Li, Guoyou Qin, Zeyan Liew, Jing Hu, Krisztina D. László, Fangbiao Tao, Carsten Obel, Jørn Olsen, Yongfu Yu.

**Project administration:** Jiong Li, Guoyou Qin, Yongfu Yu.

**Resources:** Jiong Li, Guoyou Qin, Yongfu Yu.

**Software:** Jiong Li, Guoyou Qin, Yongfu Yu.

**Supervision:** Jiong Li, Guoyou Qin, Yongfu Yu.

**Validation:** Jiong Li, Guoyou Qin, Yongfu Yu.

**Visualization:** Jiong Li, Guoyou Qin, Yongfu Yu.

**Writing – original draft:** Chen Huang, Jiong Li, Guoyou Qin, Yongfu Yu.

**Writing – review & editing:** Chen Huang, Jiong Li, Guoyou Qin, Zeyan Liew, Jing Hu, Krisztina D. László, Fangbiao Tao, Carsten Obel, Jørn Olsen, Yongfu Yu.

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
