## [Editor Report · Decision Letter 0]

15 Feb 2021

Dear Dr Yu, 

Thank you for submitting your manuscript entitled "Maternal hypertensive disorder of pregnancy and offspring early-onset cardiovascular disease in childhood, adolescence, and young adulthood: A national population-based cohort study" for consideration by PLOS Medicine for our upcoming Special Issue.

Your manuscript has now been evaluated by the PLOS Medicine editorial staff as well as by the Guest Editors, and I am writing to let you know that we would like to send your submission out for external assessment.

Once your full submission is complete, your paper will undergo a series of checks in preparation for external assessment. 

Kind regards,

Richard Turner, PhD

rturner@plos.org

---

## [Decision Letter · Decision Letter 1]

23 Apr 2021

Dear Dr. Yu,

Thank you very much for submitting your manuscript "Maternal hypertensive disorder of pregnancy and offspring early-onset cardiovascular disease in childhood, adolescence, and young adulthood: A national population-based cohort study" (PMEDICINE-D-21-00753R1) for consideration at PLOS Medicine for our upcoming Special Issue. 

Your paper was evaluated by the Guest Editors for the issue, discussed among the editorial team and sent to independent reviewers, including a statistical reviewer. The reviews are appended at the bottom of this email and any accompanying reviewer attachments can be seen via the link below:

[LINK]

In light of these reviews, we will not be able to accept the manuscript for publication in the journal in its current form, but we would like to invite you to submit a revised version that addresses the reviewers' and editors' comments fully. You will appreciate that we cannot make a decision about publication until we have seen the revised manuscript and your response. We expect to seek re-review by one or more of the reviewers, and may enlist an additional reviewer.

We hope to receive your revised manuscript by May 14 2021 11:59PM. Please email us (plosmedicine@plos.org) if you have any questions or concerns.

Please let me know if you have any questions, and we look forward to receiving your revised manuscript. 

Sincerely,

Richard Turner, PhD

rturner@plos.org

In the abstract, please quote summary demographic details for study participants.

Please add a new final sentence to the "Methods and findings" subsection of your abstract, which should begin "Study limitations include ..." or similar and quote 2-3 of the study's main limitations. 

Please add bullets to the individual points in your author summary. 

Early in the Methods section, please state whether the study had a protocol or prespecified analysis plan, and if so attach the relevant document(s) as supplementary files, referred to in the text. Please highlight non-prespecified analyses. 

At line 186, would " ... preplanned sensitivity analyses:" be appropriate?

At line 330 and any other relevant points, please make that "... also worth mentioning.".

Throughout the text, please remove spaces from within your reference call-outs (e.g., "... undeveloped countries [3,4].").

At the end of the main text, please remove information on funding, competing interests and data availability. In the event of publication, this information will appear in the article metadata, via entries in the submission form. 

In the reference list, please remove the information on competing interests and so on from reference 4, and any other relevant references. 

Noting reference 26, please ensure that all references have full access information.

Noting table 1, please substitute "sex" for "gender" where appropriate. 

Please move figure S1 to the main body of the paper. 

Please add a completed checklist for the most appropriate reporting guideline, e.g., STROBE, labelled "S1_STROBE_Checklist" or similar and referred to as such in your Methods section. In the checklist, please refer to individual items by section (e.g., "Methods") and paragraph number rather than by line or page numbers, as the latter generally change in the event of publication, 

Comments from the reviewers:

*** Reviewer #1: 

Using Danish national health registries, this population-based cohort study aims to examine the association between maternal HDP and early-onset CVD in offspring from birth to young adulthood (up to 40 years), and whether co-existing maternal history of CVD and diabetes further increases the risk of CVD among offspring. 

Comments:

This is a thorough and comprehensive analysis, applying an appropriate and rigorous statistical methodology.

An extensive array of sensitivity and subgroup analyses have been completed demonstrating robustness and providing confidence in the results.

The authors have also adequately acknowledged the main study limitations within the discussion section.

*** Reviewer #2: 

This is a very interesting study involving a very large number of subjects exposed to a variety of hypertensive disorders of pregnancy including preeclampsia and eclampsia. The authors also divide preeclampsia in early onset and late onset. The authors conclude that exposure to preeclampsia is associated with a variety of adverse cardiovascular events over a broad range of ages;up to the age of 40 years. The authors also report that maternal history of diabetes and cardiovascular disease are risk factors for cardiovascular disease in the offspring. The data are robust, the analysis very detail and I suggest that the manuscript gets published in its current form

*** Reviewer #3: 

Maternal hypertensive disorder of pregnancy and offspring early-onset cardiovascular disease in childhood, and young adulthood:A national population-based cohort study. by Huang, Li, Qin, Liew, Hu et al.

This is a good study and gives unique information from a large population registry in Denmark. There were only a few issue that concerned me and may require the authors to remove of explain.

 the conclusion that treatment of the ?maternal hypertensive disorder of pregnancy may reduce the burden of CVD in offspring is unfounded and cannot be stated and should be removed.

 I have a few questions:

 ?how many in the cohort were lost to follow-up due to emigration and how would this confound the results due to lost data?

 the CVD endpoints are very inclusive and include both arterial and venous disease. Although these may all be linked to disorders of coagulation, they have many other different risk factors during life and this may act as a confounder. How can this be managed differentially?

 rheumatic heart disease was included as a CVD. This disorder is an infective disease (streptococcus) with a genetic risk factor component very different to the other CVD studied. I note there were only a few patients in this group, but I wonder why this group was included?

 In the discussion, inclusion of smoking as a potentially lifestyle risk factor in the cohort should be mentioned alongside physical activity and alcohol use as smoking is one of the "big five" CVD risk factors.

I would keen for this study to be published as it adds useful information to the study of maternal placental syndromes.

***

[LINK]

---

## [Decision Letter · Decision Letter 2]

6 Sep 2021

Dear Dr. Yu,

Thank you very much for re-submitting your manuscript "Maternal hypertensive disorder of pregnancy and offspring early-onset cardiovascular disease in childhood, adolescence, and young adulthood: A national population-based cohort study" (PMEDICINE-D-21-00753R2) for consideration at PLOS Medicine for our upcoming Special Issue. We apologize for the delay in sending you a decision. 

I have discussed the paper with the guest editors and it was also seen by one reviewer. I am pleased to tell you that, provided the remaining editorial and production issues are fully dealt with, we expect to be able to accept the paper for publication in the journal.

[LINK]

Please let me know if you have any questions, and we look forward to receiving the revised manuscript.   

Sincerely,

Richard Turner, PhD

rturner@plos.org

Requests from Editors:

In the abstract and throughout the results section, please quote p values alongside 95% CI, where available.

At line 62, please make that "... an increased risk of early onset CVD ..." (the "23%" is implicit in the quoted HR).

At line 73, please make that "... including smoking ..." and remove "etc.". 

At line 88, please make that "extent".

At line 95, we suggest making that "(67% increased risk)" for clarity. 

Thank you for including the study protocol. Were there any non-prespecified analyses? 

At lines 230 and 232, the upper bound of the 95% CI is quoted as "1.26" and "1.27". Please check that both numbers are correct.

Please revisit the author list for reference 28 so that the format matches that of the other references. 

We suggest breaking the study protocol and STROBE checklist out into separate attached files, labelled and referred to in the text as "S2_STROBE_Checklist" and similar. 

Comments from Reviewers:

*** Reviewer #4: 

Review of:

Maternal hypertensive disorder of pregnancy and offspring early-onset cardiovascular disease in childhood, adolescence, and young adulthood: A national population-based cohort study

This study set out to elucidate the association between maternal hypertensive diseases of pregnancy and early-onset CVD in childhood and youth (before 40 years of age). It was conducted as a population-based cohort study of 2.5 million individuals. The study found that maternal HDP (defined as preeclampsia, eclampsia, gestational hypertension or pre-existing hypertension) increased the risk of early CVD by 23%. Specific risk increases were found for MI and hypertension. The association was strongest for early-onset and severe preeclampsia with a 48% risk increase for CVD.

The study however lacked information on some potential confounders such smoking and physical activity, yet a range of other covariables were available.

Major Comments:

This study is unique and has beautiful statistical strength in having such a vast register-based material with follow up over several decades. I congratulate the authors on this feat. Further extensive subgroup analyses have been performed including sibship analyses, timing of delivery and stratified analyses by offspring sex, singleton/gemelli, maternal parity and a range of maternal covariables.

This is a rather extraordinary study which warrants immediate publication and follow-up studies. 

Minor comments:

Perhaps I lack understanding of the nature of sibship analysis, but I believe maybe some wording is missing from lines 242-247, to fully understand the implications of the analyses?

***

[LINK]

---

## [Editor Report · Decision Letter 3]

9 Sep 2021

Dear Dr Yu, 

On behalf of my colleagues and the Academic Editor, Dr Bassat, I am pleased to inform you that we have agreed to publish your manuscript "Maternal hypertensive disorder of pregnancy and offspring early-onset cardiovascular disease in childhood, adolescence, and young adulthood: A national population-based cohort study" (PMEDICINE-D-21-00753R3) in PLOS Medicine for our upcoming Special Issue.

PRESS

Sincerely, 

Richard Turner, PhD 

rturner@plos.org